# Identification of Novel Diagnostic and Prognostic Gene Signature Biomarkers for Breast Cancer Using Artificial Intelligence and Machine Learning Assisted Transcriptomics Analysis

**DOI:** 10.3390/cancers15123237

**Published:** 2023-06-18

**Authors:** Zeenat Mirza, Md Shahid Ansari, Md Shahid Iqbal, Nesar Ahmad, Nofe Alganmi, Haneen Banjar, Mohammed H. Al-Qahtani, Sajjad Karim

**Affiliations:** 1King Fahd Medical Research Center, King Abdulaziz University, Jeddah 21589, Saudi Arabia; 2Department of Medical Laboratory Science, Faculty of Applied Medical Sciences, King Abdulaziz University, Jeddah 21589, Saudi Arabia; 3Department of Clinical Data Analytics, Max Super Speciality Hospital, Saket, New Delhi 110017, India; 4Department of Statistics and Computer Applications, Tilka Manjhi Bhagalpur University, Bhagalpur 812007, India; 5Computer Science Department, Faculty of Computing and Information Technology, King Abdulaziz University, Jeddah 21589, Saudi Arabia; 6Center of Excellence in Genomic Medicine Research, King Abdulaziz University, Jeddah 21589, Saudi Arabia; 7Centre of Artificial Intelligence in Precision Medicines, King Abdulaziz University, Jeddah 21589, Saudi Arabia

**Keywords:** breast cancer, gene expression profiling, artificial intelligence, machine learning methods, diagnostic and prognostic model

## Abstract

**Simple Summary:**

Breast cancer is the most fatal female cancer, which the existing clinical and pathological information sometimes fails to diagnose accurately. Recent artificial intelligence-based studies have shown the capability of identifying molecular biomarkers using high-throughput genomics data. Our aim was to apply machine learning methods to a large cohort of transcriptomics data for gene reduction and the construction of a diagnostic model for cancer classification. Advanced statistical methods and cross-validation with another set of machine learning methods increased the accuracy of the diagnostic model and predicted a novel diagnostic nine-gene signature. Further, survival analysis revealed a novel prognostic model of eight-gene signatures. Experimental validation confirmed the expression of the identified gene signatures in breast cancer patients and increased the reliability of the study. The identified gene signature biomarkers have the potential to improve healthcare management with precise diagnosis and prognosis at a reduced cost.

**Abstract:**

Background: Breast cancer (BC) is one of the most common female cancers. Clinical and histopathological information is collectively used for diagnosis, but is often not precise. We applied machine learning (ML) methods to identify the valuable gene signature model based on differentially expressed genes (DEGs) for BC diagnosis and prognosis. Methods: A cohort of 701 samples from 11 GEO BC microarray datasets was used for the identification of significant DEGs. Seven ML methods, including RFECV-LR, RFECV-SVM, LR-L1, SVC-L1, RF, and Extra-Trees were applied for gene reduction and the construction of a diagnostic model for cancer classification. Kaplan–Meier survival analysis was performed for prognostic signature construction. The potential biomarkers were confirmed via qRT-PCR and validated by another set of ML methods including GBDT, XGBoost, AdaBoost, KNN, and MLP. Results: We identified 355 DEGs and predicted BC-associated pathways, including kinetochore metaphase signaling, PTEN, senescence, and phagosome-formation pathways. A hub of 28 DEGs and a novel diagnostic nine-gene signature (*COL10A, S100P, ADAMTS5*, *WISP1*, *COMP*, *CXCL10*, *LYVE1*, *COL11A1*, and *INHBA*) were identified using stringent filter conditions. Similarly, a novel prognostic model consisting of eight-gene signatures (*CCNE2*, *NUSAP1*, *TPX2*, *S100P*, *ITM2A*, *LIFR*, *TNXA*, and *ZBTB16*) was also identified using disease-free survival and overall survival analysis. Gene signatures were validated by another set of ML methods. Finally, qRT-PCR results confirmed the expression of the identified gene signatures in BC. Conclusion: The ML approach helped construct novel diagnostic and prognostic models based on the expression profiling of BC. The identified nine-gene signature and eight-gene signatures showed excellent potential in BC diagnosis and prognosis, respectively.

## 1. Introduction

Breast cancer (BC) is the most common cause of cancer death in women, with 1 in 8 cancer cases, and its incidence has increased significantly despite the preventive and curative approaches utilized in recent years [1,2]. In 2020, the International Agency for Research on Cancer (IARC) and World Health Organization (WHO) reported 2.26 million new BC cases and 684,996 global cases of BC mortality in females, surpassing lung cancer with 2.20 million new cases. Further, the diagnosis of new cases and BC death by 2040 is predicted to increase to over 3 million and 1 million, respectively [3,4]. BC is a heterogeneous disease and the symptoms may include a bump, skin dimpling, nipple discharge, scaly hair patch and flaky skin around the nipple, and thickness/swelling in some parts of the breast [5]. BC survival rates were found to be variable, at ~80%, ~60%, and ~40% in high-, mid-, and low-income countries, respectively.

An accurate diagnosis is key for the optimal treatment of cancer patients. At present, cancer classification and diagnosis heavily depend on the subjective evaluation of physical examination, clinical/pathological test, radiological scan, and histopathological information, but they are subject to human errors [6]. Surprisingly, medical error is the third leading cause of death, even in the most advanced countries such as the USA [7]. Additionally, in some instances, (i) incomplete or misleading clinical information, (ii) complicated radiological images, and (iii) variable, atypical, or lack of morphologic features in histological information may result in diagnostic confusion, and thus affect patient care [2].

Molecular diagnostics offer precise, fair, and efficient breast cancer classification, but are not widely applied in clinical settings. Microarray-platform-based assays, including the Affymetrix GeneChip Human Genome U133 Plus 2.0 array (Affymetrix, Santa Clara, CA, USA), have the ability to measure thousands of gene expressions simultaneously for each data point (sample) [8,9]. Expression profiling to check for variability in gene expression is an important factor influencing the precision and accuracy of clinical decisions in the diagnosis of BC [1,8,10,11,12,13]. Despite the large-scale, high-dimensional, and highly redundant type of microarray data, with numerous tools to identify genes that are differentially expressed across cancer/disease phenotypes, the interpretation of results and follow-up analysis are quite challenging. DNA-microarray-based gene expression profiling is promising for BC diagnosis and prognosis [12,14], but limitations such as small sample size, biased case vs. control distribution, multiple BC subtypes, variable populations, and different platforms, complicate the analysis, and the identification of gene signatures remains an issue [15,16,17,18,19].

In 1999, the first gene expression signature was identified to classify leukemia into acute myeloid leukemia (AML) and acute lymphoblastic leukemia (ALL). Since then, a series of gene expression signatures have been reported for various cancers to classify tumors, tumor types, tumor stages, and predict the disease prognosis [20,21].

BC is a very heterogeneous disease and is categorized into five molecular subtypes: HER2+, basal (ER−/HER2−/PR−), luminal A (ER+/HER2, with a low-proliferative phenotype), luminal B (ER+/HER2, with a high-proliferative phenotype), and normal BC [17]. Each subtype exhibits distinct transcriptomics patterns, and finding unified BC biomarkers or the gene signature applicable to all molecular subtypes remains a challenge [13,22]. Hence, multiple datasets need to be integrated to find universal diagnostic/prognostic biomarkers broadly applicable to all BC subtypes.

A stringent filtration condition drastically reduces the number of DEGs, but filters out many biologically relevant genes as well, whereas a lenient cutoff allows for many genes to pass through, and a follow-up issue arises in selecting the most interesting genes. Although conducting gene ontology and pathway enrichment analysis is useful in predicting biological processes, cellular components, molecular function, networks, and canonical pathways for the detected DEGs, the selection of the most relevant genes, a diagnostic and/or prognostic biomarker, in cancer remains a challenge. Machine learning methods and different evaluation techniques such as the Kaplan–Meier (KM) estimator might be useful in identifying the biologically relevant genes from a long DEG list, without any obvious selection way [8,11].

Another problem in high-throughput gene expression profiling is reducing the extremely high dimensionality of irrelevant or redundant gene features responsible for cancer classification accuracy. Feature selection methods have been used to select key genes from thousands of expressed genes, but the large numbers of microarray genes used in most existing methods for cancer classification often hamper the model outcomes. For an efficient diagnostic model for BC, machine-learning-based feature selection methods were applied to a smaller number of differentially expressed genes passing the standard statistical cutoff of *p* < 0.5 and log2folds change > 2 in BC.

To address these challenges, we integrated eleven GEO oligonucleotide microarray datasets to create a gene expression database of 701 samples (356 breast tumors and 345 normal breast tissues) and applied different R packages and machine learning methods on gene expression data for the molecular classification, accurate diagnosis, and prognostic evaluation of the identified gene signatures in BC. A larger sample size gave greater analysis power because it constricted the distribution of the test statistic. Further, an almost equal group (3356 vs. 3345) reduced the biases in machine-learning-based data analysis, and increases the accuracy of the model and predicted biomarkers. We also demonstrated that the combined use of molecular pathway analysis, expression analysis, feature selection methods, and survival analysis was helpful in selecting gene signatures with high confidence.

## 2. Materials and Methods

### 2.1. Data Sets and Patients

The raw gene expression data, a set of binary files in a CEL format, of BC from eleven datasets, including GSE61304, GSE42568, GSE7904, GSE3744, GSE29431, GSE26910, GSE31138, GSE71053, GSE10780, GSE30010, and GSE111662, were retrieved from the Gene Expression Omnibus (GEO) database using “GEOquery” library of the R program (https://www.ncbi.nlm.nih.gov/geo/, accessed on 2 January 2023). We selected only the human breast tumor samples to eliminate differential genetic interference in different BC cell lines. Clinicopathological information from the original studies was used for analysis. The ratio of breast tumors to normal breast was biased in the majority of the deposited GEO datasets, including GSE61304, GSE42568, GSE7904, GSE3744, and GSE26910. Thus, we included additional normal breast cases (GSE30010 and GSE111662) to balance the data (n = 701, 356 = BT vs. 345 = NB) for a better outcome, while identifying DEGs or applying ML methods to develop a diagnostic model (Table 1). This study was approved by the university’s CEGMR bioethical committee (16-CEGMR-bioeth-2022), dated 13 October 2022, and we recruited patients for the validation of potential biomarkers after obtaining their consent.

### 2.2. Preprocessing and Differential Expression Analysis

The median expression values of less than 5.55 intensity on the log2 scale of each probe, indicating the failure of true hybridization, were filtered out. We also excluded the probes expressed in less than two samples. We merged all the raw CEL files (n = 701) and applied the RMA method for the normalization of expression values, and generated box plots using the “oligo” package from R software. Principal component analysis (PCA) was performed using “prcomp function”, and hierarchical clustering was carried out using the “pheatmap” R package to correlate the samples with the probes.

We used linear models for the microarray “Limma” package of R to identify differentially expressed genes (DEGs), using an empirical Bayesian method to assess the differences in gene expression. Wang et al. (2021) conducted a study demonstrating the superior performance of the moderated *t*-test when the sample size was ≥40. [23,24]. Our study, with a sample size of 701, comprising 356 breast tumors and 345 normal breast samples, exceeds the required 95% power of the test, and the nearly equal representation of the test (tumor) and control (normal) groups mitigates biases in machine-learning-based data analysis, and enhances the accuracy of the model and predicted biomarkers. The “decideTests” function was used to differentiate between the altered (up or down) and normal expression. The “topTable” function from R was applied with a cut-off-adjusted *p*-value (Benjamini–Hochberg-corrected false discovery rate)  < 0.05 and log2 fold change > ±2 to detect the most significant DEGs in BC compared with normal samples. Unannotated probes, not representing genes, were removed, and duplicate probes, representing single genes, were averaged for expression values to get a unique set of DEGs.

### 2.3. Functional Pathway and Gene Set Enrichment Analysis

We used a comprehensive set of functional annotation tools, such as the QIAGEN Ingenuity Pathway Analysis (IPA knowledgebase v84978992, QIAGEN, USA) and WEB-based Gene SeT AnaLysis Toolkit (WebGestalt 2019, https://www.webgestalt.org/, accessed on 2 January 2023), to investigate and understand the biological meaning of long-list significant DEGs [1,25,26]. We explored gene ontologies, enriched and canonical pathways, upstream regulators, disease and functions, and the networks associated with BC. Over-representation (or enrichment) analysis (ORA), a statistical method, was used to determine the presence of known genes in pre-defined sets, as well as in dataset/DEGs.

### 2.4. Machine Learning and Feature Selection Methods

We applied machine learning methods to BC transcriptomics data and considered performance measurements such as classification accuracy, specificity, sensitivity, and AUC, to identify the most informative features. To determine the effectiveness of a classification model, a set of performance metrics was used for assessment, such as measuring the model’s ability to accurately classify instances into the correct categories. We used the confusion matrix to compute the accuracy, precision, recall, and F1 score, as shown in Equations (1)–(4).
(1)$$Accuracy=\frac{TP+TN}{TP+FP+FN+TN}$$
(2)$$Precision=\frac{\sum TP}{\sum TP+\sum FP}$$
(3)$$Recall=\frac{\sum TP}{\sum TP+\sum FN}$$
(4)$$F1-score=\frac{2*precision*recall}{precision+recall}$$

In addition, the model’s performance was evaluated by plotting the receiver operating characteristic curve (ROC) and the area under the ROC (AUC), which is a metric used to measure the model’s effectiveness. Models with larger AUCs are considered to have higher performance.

The Scikit-learn (sklearn) in python platform was used to build the ROC curves of the DEGs and measure the AUC to compare the diagnostic value of the DEGs, and to predict the accuracy of the detected DEGs. The ROC curve, reflecting the relationship between sensitivity and specificity, and AUC were used to determine the diagnostic value of a factor in a specific disease, with AUC values between 0.5 and 1 representing low and high authenticity, respectively.

We used seven machine learning algorithms, including (i) recursive feature elimination with cross validation (RFECV) with logistic regression, (ii) RFECV with support vector machine (SVM), (iii) Lasso regularization (L1) with logistic regression, (iv) Lasso regularization (L1) with support vector classification (SVC-L1), (v) random forest (RF) classifier, (vi) extremely randomized trees (extra trees) classifier, and (vii) genetic algorisms (GA), to find the most significantly expressed genes in all samples (n = 701, 356BT + 345NB) and construct the diagnostic model from candidate DEGs. To validate the constructed diagnostic and prognostic models, we used five additional ML methods, including (i) adaptive boosting (AdaBoost), (ii) gradient-boosted decision trees (GBDT), (iii) K-nearest neighbors (KNN), (iv) multilayer perceptron (MLP), and (v) the extreme gradient boosting (XGBoost).

#### 2.4.1. RFECV with Logistic Regression or with SVM

We used RFECV with logistic regression and SVM in Python using the RFECV class from the scikit-learn library. The RFECV with logistic regression or SVM is a method of feature selection in machine learning. It is a combination of two techniques: recursive feature elimination (RFE) and cross validation (CV). RFE is a backward selection algorithm that starts with all the features and removes the weakest feature until a specified number of features is left. CV, on the other hand, is a technique used to evaluate the performance of a model by dividing the data into several folds and training the model on different folds, while testing it on one fold at a time. In RFECV with logistic regression or SVM, the RFE algorithm is combined with CV to eliminate features, while also evaluating the performance of the logistic regression model. This helps determine the optimal number of features that provide the best performance, while avoiding overfitting. By combining these two techniques, RFECV with logistic regression or SVM ensures that the final feature set is not only informative, but also generalizable to new data.

#### 2.4.2. LASSO Regularization (L1) Using Logistic Regression or Support Vector Classification

We used L1 with logistic regression and SVM from the scikit-learn library to identify the most significant genes [27]. LASSO regularization is a technique used to reduce the number of features in a model and prevent overfitting. The technique shrinks the magnitude of the coefficients using a penalty term proportional to the absolute value of the coefficients, resulting in some coefficients becoming zero. It helps select the most relevant features, while reducing the impact of irrelevant or noisy features on the model’s performance. L1 regularization is particularly useful when dealing with a large number of features or highly correlated features. The loss function of Lasso regression is defined as shown in (5):(5)$$\overset{n}{\underset{i=1}{\sum }}(Y_{i}-\overset{p}{\underset{j=1}{\sum }}X_{ij}\beta_{j}{)}^{2}+\lambda \overset{p}{\underset{j=1}{\sum }}\left|\beta_{j}\right|$$
where lambda is the regularization parameter that controls the strength of the penalty term.

In logistic regression with LASSO regularization, the L1 penalty term helps reduce the impact of irrelevant or noisy features on the model’s performance by shrinking their coefficients toward zero. This can improve the model’s interpretability and reduce the risk of overfitting, especially when dealing with high-dimensional data.

In SVM, LASSO regularization is implemented by adding a penalty term to the objective function that minimizes the classification error. The penalty term is dependent on the magnitude of the coefficients of the features, so larger coefficients receive a larger penalty. This ensures that features with a large impact on the classification result receive a smaller penalty and are more likely to be included in the final model.

#### 2.4.3. Random Forest

The random forest classifier is a machine learning algorithm used for both classification and regression problems [28]. It is an ensemble of decision trees, where each tree is trained on a random subset of data. The final prediction is made by taking the average of all the trees’ predictions. In feature selection, a random forest classifier is used to select the most important features in the dataset. The algorithm calculates the importance of each feature by measuring the average decrease in impurity for that feature. The higher the average decrease, the more important the feature is considered.

#### 2.4.4. Extra Trees Classifier

The extra trees classifier is an ensemble machine learning algorithm that can be used for feature selection in Python [29]. It is a type of random forest classifier where multiple decision trees are grown and combined to make a prediction. The algorithm works by randomly selecting a subset of features at each split in the tree and determining the most important features based on their impact on the final prediction. By aggregating the feature importance scores across all trees, the extra trees classifier can provide a ranking of the most important features for a given dataset. This can be useful in identifying the most relevant features for building a predictive model and reducing the dimensionality of the data.

#### 2.4.5. Genetic Algorithm

This algorithm uses principles of evolution and natural selection to find the optimal combination of features that result in the best model performance. The algorithm starts with a random set of features and uses a fitness function to evaluate the performance of each combination. The best performing combinations are then recombined and mutated to create a new generation of features, and the process continues until a satisfactory set of features is found [30].

#### 2.4.6. XGBoost

This algorithm is an ensemble learning method that works by combining multiple weak models into a strong one [31]. It uses gradient boosting, which is a method of iteratively training decision trees on residuals to improve the model performance. It provides faster computation and parallelization of training, which is useful when working with large datasets. It also has built-in regularization techniques to reduce overfitting, which is a common problem in machine learning.

#### 2.4.7. GBDT

This is a machine learning algorithm that works by building an ensemble of decision trees in a way that each subsequent tree focuses on the errors made by the previous trees [32]. This iterative process results in a model that can learn complex non-linear relationships in data. It has the ability to handle large datasets, handle missing data, and provide accurate predictions with high interpretability.

#### 2.4.8. MLP

This is a type of feedforward artificial neural network that consists of multiple layers of nodes that process information from the input layer to the output layer through a series of nonlinear transformations [33]. The nodes in each layer are connected to the nodes in the previous and next layers, and each node applies an activation function to the weighted sum of its inputs. The goal of training the MLP is to minimize this cost function by adjusting the weights and biases in the network using an optimization algorithm such as gradient descent. This allows the model to learn the best set of weights that can accurately predict the binary classification labels for unseen data. The cost function in the binary classification of MLP uses the binary cross-entropy loss of function and is defined as shown in (6):(6)$$cross-entropy=-\frac{1}{N}\overset{N}{\underset{i=1}{\sum }}\left(y_{i}\log(p_{i\;})+\left(1-y_{i}\right)\log(1-p_{i\;})\right)$$
where *N* is the number of samples, 

$y_{i}\;$ is the actual outcome, 

$p_{i}$ is the probability of the tumor class, and

$1-p_{i}\;$ is the probability of the normal class.

#### 2.4.9. AdaBoost

This is an ensemble learning algorithm that combines several weak learners to create a strong learner [34]. It works by repeatedly fitting a weak learner to the data, and adjusting the weights of the training samples to focus on the misclassified ones. The algorithm then combines these weak learners to form a strong learner that is capable of accurately predicting the target variable.

#### 2.4.10. KNN

The K-nearest neighbors (KNN) is a popular machine learning algorithm belonging to the family of instance-based or lazy learning algorithms, which means that it does not attempt to learn a function from the training data [35]. Instead, KNN stores all the training data, and classifies new data based on the similarity of its features to those in the training set. The number of nearest neighbors (K) is a hyperparameter that can be tuned to improve performance.

### 2.5. Survival Analysis Using the Kaplan–Meier Estimator

The KM estimator, a statistical technique tool (available at https://kmplot.com/analysis/, accessed on 10 February 2023) was used for calculating survival probability functions to investigate the overall and relapse-free survival of prognostic genes for breast cancer patients. It is assumed that the occurrence of the event is fixed in time, and both the censored observations and data points have an equal chance of survival [8,36].

The mathematical expression of KM is expressed as shown in (7):(7)$$S\left(t\right)=\underset{i:t_{i}\leq t}{\prod }\frac{n_{i}-d_{i}}{n_{i}}=\underset{i:t_{i}\leq t}{\prod }\left(1-\frac{d_{i}}{n_{i}}\right)$$
$S\left(t\right)\;$ stands for survival function.

In this context, $n_{i}\;$ refers to the count of individuals at risk at a specific time *tᵢ*, and $d_{i}\;$ is the count of events that happen at the same time, *tᵢ*. The survival curve remains unchanging between the two events or times, i.e., between *tᵢ* and *tᵢ* + 1 [36].

This analysis was conducted for mRNA (gene chip) microarray data for the relapse-free and overall survival. The KM analysis was performed with a confidence interval and log rank *p*-value cut-off of >95% and ≤0.05, respectively. We proceeded to further check the mRNA (RNA seq) datasets for overall survival for genes which were significant in both microarray data for relapse-free survival and overall survival. Finally, we established eight gene hubs (four upregulated and four downregulated) for prognostic importance.

The web-based KMplot tool incorporates three databases in the background: TCGA, EGA, and GEO [37]. The Kaplan–Meier method is a strong non-parametric statistical approach used for predicting the likelihood of survival. The KM analysis was performed with a confidence interval and log rank *p*-value cut-off of >95% and ≤0.05, respectively.

### 2.6. RNA Isolation and qRT-PCR

Trizol was used to lyse the cells, and chloroform and isopropanol were used to extract RNA. After determining the RNA concentration, the cDNA (complimentary deoxyribonucleic acid) was reverse-transcribed. The primer sets were designed for the identified gene signature using Primer-3 software (V.0.4.0). ABI 7500 instruments were used for real-time quantitative PCR. Endogenous *GAPDH* gene expression was measured as the internal control to determine the relative expression of the detected genes. The reaction was run in a final volume of 10 μL, comprising 5 μL SYBR-Green qPCR master mix (KAPA Biosystems, Wilmington, MA, USA), 10 pmol of each primer, and 20 ng genomic DNA. PCR was performed in triplicate using the SYBR-Green qPCR master mix (KAPA Biosystems, USA) in a 96-well plate. Raw data were generated through the use of StepOne Plus™ Real-Time PCR Systems and Data Assist software. qPCR data were analyzed by ∆∆CT or the Livak method, and the GraphPad PRISM software was used for presentation.

### 2.7. Statistical Analysis

All statistical analyses were conducted using R software (version v.4.2.2) (R core team 2021). R was also used for the picture generation. The chi-squared test was used to compare categorical variables of patient characteristics. The Wilcoxon rank sum test was used to compare the expression signature. The *p*-values were adjusted for multiple comparisons using the Benjamini–Hochberg method, and the default value <0.05 was considered statistically significant, otherwise specified. Cox regression analysis (univariate and/or multivariate) was used to assess the contribution of all parameters, such as evaluating the independent predictive OS performance of different clinical factors and the detected biomarkers. The KM curve and time-dependent ROC curve were drawn by the R package “survminer” and “survivalROC”, respectively.

## 3. Results

### 3.1. Differentially Expressed Genes in BC

Breast tumor (356) and normal breast tissue (345) samples from the Affymetrix GeneChip Human Genome U133 Plus 2.0 arrays platform (HG-U133_Plus_2) with 54,675 features/probes from 11 different GEO data series were used as discovery cohorts. Out of 54,675 hybridized probes, only 46,597 probes passed the cut-off: median expression >5.5 and present in at least two samples. The expression data were GC-RMA-normalized. The raw intensities and RMA-normalized expression values were shown in Boxplot (Figure 1).

For a tumor-normal matrix, the “decideTests” function differentiated 46,597 probe signal intensities into altered (over-expressed (15,000), under-expressed (21,952)) and unaltered (9645) expression. The “topTable” functions revealed the most significant DEGs in breast cancer (n = 487) for the screening criteria of the adjusted P-value (Benjamini–Hochberg-corrected false discovery rate)  < 0.05 and fold change (log2FC > ±2). Additionally, unannotated probes, not representing valid genes (n = 20), were removed first, then duplicate genes, multiple probes representing single genes (n = 193), were averaged for expression values (n = 83) to get a unique set of DEGs (n = 355, upregulated = 77 and downregulated = 278) (Figure 2 and Table 2).

### 3.2. Function Pathway Analysis and Network Enrichment Analysis

Functional analysis based on the Z-score and −log(*p*-value) indicated activation of the kinetochore metaphase signaling pathway (2.71, 7.35), PTEN pathway (2.64, 2.31), HOTAIR regulatory pathway (2.12, 2.71), and WNT/β-catenin signaling pathway (2.45, 1.47), and suppression of the senescence pathway (−3.16, 3.02), phagosome formation (−3.15, 2.09), FAK signaling (−3.128, 1.74), oxytocin signaling pathway (−3.05, 3.8), and breast cancer regulation by Stathmin1 (−3.0, 2.06) (Figure 3). Most significantly enriched molecular processes were the extracellular matrix, cell division, mitotic cell cycle process, cell migration, and the regulation of cell proliferation (Table 3).

### 3.3. Machine Learning Algorithms for the Identification of Diagnostic Biomarker Genes

Initially, seven ML algorithms predicted genes with diagnostic importance using 355 DEGs significantly expressed in BC samples, and important genes predicted by at least four ML models were selected for further analysis (n = 65). Additionally, we analyzed each dataset individually and checked the status of 355 DEGs in each dataset; the genes present in at least four datasets were selected for further analysis (n = 94). We identified 28 common genes passing both criteria (DEGs > 3 ML and DEGs > 4 datasets) as the potential hub of BC diagnostic and prognostic biomarkers (Figure 4A). With more stringent conditions (DEGs > 5 ML and DEGs > 7 dataset) and based on their role in tumorigenesis, a novel diagnostic nine-gene signature (*COL10A, S100P, ADAMTS5, WISP1, COMP, CXCL10, LYVE1, COL11A1,* and *INHBA*) was identified for BC. An unsupervised hierarchical clustering-based heatmap showed a correlation in a pairwise fashion between the samples and probes. A heatmap of unfiltered probes (n = 54676) was ambiguous and non-conclusive, while the unique set of DEGs (n = 355), hub genes (n = 28), and gene signatures (n = 9) for BC diagnosis showed a distinct correlation between the samples and gene expression (Figure 4B).

### 3.4. Machine-Learning-Algorithm-Based 10-Fold Cross-Validation

We used a 10-fold cross-validation technique to evaluate the performance of ML models for diagnostic and prognostic gene signatures. To perform 10-fold cross-validation, the dataset was divided into 10 equally sized folds. The model was then trained and validated 10 times, each time using a different fold for validation and the remaining nine folds for training. The process was repeated for all the folds, and the results were averaged to obtain an estimate of the model’s performance. This provided a more reliable estimate of the model’s performance compared to using a single-train test split, which may be biased based on the specific data that were selected; it can also help prevent overfitting.

For validating the diagnostic performance of our nine-gene signature, a new set of ML algorithms (GBDT, XGBoost, AdaBoost, KNN, and MLP) was employed to evaluate the model. By comparing the diagnostic efficiency, accuracy, and precision of different algorithms, the constructed diagnostic model was validated (Figure 5 and Table 4). Each ML model’s performance was evaluated by measuring a range of performance metrices, including AUC, accuracy, precision, recall, and the F1 score. Here, all the ML methods predicted were above 95, and the ML model had the greatest AUC value indicating the candidate gene signature as a potential biomarker. KNN showed the highest values for all the evaluation metrics (mean F1 = 0.982), which indicates that it performed the best among the five models. In contrast, MLP showed the lowest values for all evaluation metrics. Furthermore, biomarkers that could distinguish disease samples and normal samples were analyzed according to PCA, as it groups the samples based on similarities (Figure 6).

### 3.5. Survival Analysis to Identify Genes with Prognostic Importance

Survival analysis using the KM estimator was conducted for 28 hub genes. First, relapse-free survival and overall survival analyses were performed for mRNA (gene-chip), followed by the overall survival for mRNA (RNA seq). We identified a novel prognostic gene signature of eight genes (*CCNE2, NUSAP1, TPX2, S100P, ITM2A, LIFR, TNXA,* and *ZBTB16*) that were significant (log-rank *p*-value < 0.05) under both RFS and OS conditions (Table 5 and Table 6, and Figure 7 and Figure 8). The hazard ratio (HR) compares the risk of death (overall survival) and postoperative follow-up (relapse-free survival) occurring between the high and low expressions of the gene. The HR value < 1 or >1 indicates that the risks associated with a lower expression and higher expression of the gene are significantly different. On the other hand, the confidence intervals (CI) indicate the level of uncertainty around the estimated survival probability at each time point. A narrower confidence interval indicates that the survival estimate is more precise and that the sample size is large enough to produce reliable results. The CI value demonstrates the precision and reliability of the results. Finally, the log-rank *p*-value measures the statistical significance that helps determine whether the observed difference in survival between the groups (high and low expression of the gene) is statistically significant (<0.05), i.e., unlikely to have occurred by chance. Therefore, log-rank *p*-values are the deciding factors for survival significance.

First, we validated the prognostic eight-gene signature using mRNA (RNA seq) dataset based on the RFS and OS analyses by collectively measuring the hazard ratio (HR), confidence interval (CI), and log-rank *p*-value (Figure 9 and Table 7), and the prognostic signatures were again validated using five ML methods, including GBDT, XGBoost, AdaBoost, KNN, and MLP. Among the five ML models, GBDT showed the highest values for the mean AUC (0.993), mean accuracy (0.980), and mean F1 score (0.98), while XGBoost showed the highest mean precision (0.981). KNN showed the second-highest values for all the evaluation metrics, while MLP showed the lowest values (Figure 10 and Table 8).

### 3.6. qRT-PCR Analysis

qPCR was used to confirm the identified BC biomarkers (gene signatures) by determining the relative expression of 16 genes (nine-gene signature for diagnosis: *COL10A*, *S100P*, *ADAMTS5*, *WISP1*, *COMP*, *CXCL10*, *LYVE1*, *COL11A1*, and *INHBA*; and eight-gene signature for prognosis: *CCNE2*, *NUSAP1*, *TPX2*, *S100P*, *ITM2A*, *LIFR*, *TNXA*, and *ZBTB16*), with an overlap of the *S100P* gene (Figure 11). *GAPDH* was used as the internal control. We found qPCR results in concordance with microarray-analyzed expression patterns (Table 9).

## 4. Discussion

In recent years, multiple molecular diagnostic prognostic and predictive biomarkers have been proposed, and despite the availability of few molecular tests, traditional pathological factors such as the number of lymph node metastases, tumor size, and tumor grade, continue to be mandatory for clinical decisions [38]. However, in the era of personalized treatment, these factors alone are inadequate and require molecular/genomic assistance, as cancer occurs via genetic alterations that transform normal cells into tumor cells. Although significant knowledge exists related to carcinogenesis, a complete understanding of cancer development mechanisms is still required. In recent years, genomics and proteomics have played a vital part in the development of different biomarkers for breast cancer [39,40]. Gene expression profiling can detect genetic alterations in the origin, growth, proliferation, and metastasis of tumors, and classify them accordingly. Gene expression signatures, derived from DEGs, specifically correlate these genetic alterations with clinical variables such as the diagnosis and prognosis [20,41]. A correct and timely diagnosis is the starting point of treatment and determination of prognosis is the most immediate challenge in patient management. This can be best achieved through a combination of traditional clinicopathological prognostic factors, molecular biomarkers such as single-gene tests (ER, PR, HER2) and specific multigene tests (gene signatures). 

Among the 355 DEGs identified in a combined BC cohort, *COL11A1, TOP2A, S100P, COL10A1,* and *RRM2* were the most upregulated, while *ADH1B, ADIPOQ, PLIN1, LEP,* and *LPL* were the most downregulated DEGs in BC. The pathway and enrichment analyses of DEGs revealed activation of the kinetochore metaphase signaling pathway, PTEN pathway, HOTAIR regulatory pathway, etc., and suppression of the senescence pathway and phagosome formation pathways in BC. The most significantly enriched molecular processes were the extracellular matrix, cell division, mitotic cell cycle process, cell migration, and regulation of cell proliferation. First, to verify the reliability of our method of screening for biomarkers, we confirmed our finding of DEGs, pathways, and gene ontologies using literature mining and verification. Matching our results with previous findings was good evidence that they are indeed involved in the development and progression of BC [42,43,44,45,46].

Kinetochore architecture and its functional regulation is one of the most fascinating multi-protein machineries in a cell [47]. The kinetochore metaphase signaling is essential for chromosome segregation in mitosis and meiosis [48]. The critical regulators of alignment and segregation of chromosomes during mitosis, aurora B kinase (AURKB), dual specificity protein kinase TTK (Mps1), and kinetochore protein NDC80 homolog (NDC80) previously reported were significant in our study too [49]. Another essential pathway that was significantly upregulated in our study was PTEN/PI3K/AKT. This controls the signaling of numerous biological processes, including apoptosis, cell proliferation, cell growth, and metabolism. Phosphatase and tensin homolog deleted on chromosome 10 (PTEN) is a dual protein/lipid phosphatase, of which the main substrate is phosphatidyl-inositol,3,4,5 triphosphate (PIP3), the product of PI3K [50,51]. The PTEN tumor suppressor is the chief brake of the PI3K-Akt pathway and a common target for inactivation in somatic cancers [52]. PTEN activity is frequently lost in several metastatic human cancers due to mutations, deletions, or promoter methylation silencing [50]. Senescence is associated with mitochondrial metabolic activities such as the tricarboxylic acid cycle, oxidative phosphorylation, and glycolytic pathways. The old senescent cells die during aging or apoptosis. The senescence pathway promotes cell cycle arrest triggered in response to stress with increased AMP/ADP:ATP and NAD^+^/NADH ratios, and activating AMPK, p53, p16, KRAS, etc. [53,54,55]. The in vitro demonstration of oncogene-induced senescence establishes senescence as a vital tumor-suppressive mechanism, in addition to apoptosis. Senescence not only stops the proliferation of premalignant cells (tumorigenesis), but also eases the clearance of affected cells through the immunosurveillance [56]. In vivo studies showed that suppression of the senescence pathway can also promote mammary tumorigenesis [57].

AI and ML techniques based on automated medical diagnosis are increasing gradually for clinical, pathological, and radiological reports. The fusion of multiple techniques in different types of data processing for cancer study must be a further instrument to obtain successful results. An earlier convolution neural network approach had been applied for image processing in medical diagnosis [58]. However, using AI and ML in the evaluation of high-throughput genomics data from patients in diagnostic decision-making is still a bottle neck in healthcare [59,60,61]. Typically, microarray data have thousands of features (genes/probes), but only a few samples (in tens or hundreds). For ML classification, it is better to have a large cohort with fewer features. Eleven BC datasets from different studies were integrated to increase the cohort size. Transcriptomics profiling resulted in 355 DEGs associated with BC, but this number was technically too big to recommend for gene signature biomarkers for diagnostic or prognostic tests. However, AI and ML have the potential to filter out genes with the best diagnostic and prognostic importance. Thus, for BC diagnosis via binary classification (whether or not BC), we used seven ML and feature selection methods (RFECV-LR, RFECV-SVM, RF, extra trees, LASSO, SVM-L1, SVM-L2, and GA) for gene reduction, and found high accuracy in the models. We identified a hub of 28 genes predicted by at least three ML methods and present in at least four BC datasets. RFECV-LR and RFECV-SVM improved the classification accuracy of logistic regression by selecting the most relevant features for the model and helped reduce overfitting by removing irrelevant or redundant features. Recursive feature elimination was utilized to rank the genes, with a random forest classifier used to evaluate gene fitness through five-fold cross-validation [62]. The extra trees method was versatile, less prone to overfitting, computationally efficient, robust to noise, and could handle missing data [29,63]. LASSO had advantages in its ability to handle multicollinearity, and provided a sparse solution for both variable selection and shrinkage problems [64,65]. SVM models were frequently used for classification and regression tasks using L1 and L2 regularization [66]. L1 regularization had improved the interpretability of the model and reduced overfitting by encouraging sparsity and selecting only the most relevant features for the classification task. GA utilized the concept of survival of the fittest and was based on a population-based search approach for a robust and efficient search [67].

Based on the importance of the 28 hub genes in BC and using stringent filter conditions such as the genes predicted to be diagnostically important by at least five ML methods and present in at least seven BC datasets, a novel nine-gene signature (*COL10A*, *S100P*, *ADAMTS5*, *WISP1*, *COMP*, *CXCL10*, *LYVE1*, *COL11A1*, and *INHBA*) was identified. Similarly, by evaluating 28 hub genes using RFS and OS analyses by KM plot, a novel prognostic model consisting of an eight-gene signature (*CCNE2*, *NUSAP1*, *TPX2*, *S100P*, *ITM2A*, *LIFR*, *TNXA*, and *ZBTB16*) was identified. Many gene expression signatures have been proposed for BC diagnosis and prognosis in recent years; few are under trial and five of them succeeded to get FDA approval for commercial and clinical application, including OncotypeDX (21-gene signature), MammaPrint (70-gene signature), Prosigna (58-gene signature), EndoPredict (12-gene signature), and Breast Cancer Index (7-gene signature) [68,69].

Gene signature validation was crucial before recommendation for further analysis and clinical trials. We used 10-fold-cross validation by five ML methods including KNN, GBDT, AdaBoost, XGBoost, and MLP. Several studies have reported successful applications of these ML methods in BC gene hub/signature validation [70,71,72,73,74,75]. KNN-based validation was used to classify genes based on their expression profiles, and identify the gene clusters associated with cancer metastasis [72,73]. GBDT and AdaBoost were used to identify the key genes and pathways associated with breast cancer metastasis [72,74,76]. In addition, to identify disease-associated genes and pathways, XGBoost predicted cancer recurrence based on gene expression data [71,73]. The MLP method predicted gene clusters from expression data that were functionally related and associated with BC [70,72].

CCNE2 (Cyclin E2), involved in cell cycle regulation, can serve as an individual indicator of the likely outcome for BC patients. It is upregulated in tumor tissues and has the potential to function as a biomarker and linked to worse metastasis-free survival (MFS) outcomes and a poor overall survival [77,78]. NUSAP1 (nucleolar and spindle-associated protein 1) playing a critical role in cell division and being a useful prognostic marker, has been implicated in various types of cancer, including BC [79]. The TPX2 (targeting protein for xenopus kinesin-like protein 2), a microtubule-associated protein involved in spindle formation and cell division, is highly expressed in various cancers, including BC [80]. A high expression of TPX2 can reduce the survival time of HER2-positive patients, as well as triple negative BC [81]. S100P, a calcium-binding protein, is involved in cell proliferation, differentiation, and apoptosis. An overexpression of S100P in BC cells makes it more aggressive, and hence it has the potential as a prognostic and therapeutic biomarker [8]. ITM2A (integral membrane protein 2A) regulates cellular growth and survival, and its low expression may play a role in the progression of BC, especially at advanced stages and higher grades of triple-negative breast cancer [82]. A decreased expression of LIFR (leukemia inhibitory factor receptor) may be a marker for a poorer prognosis and reduced survival in BC [83]. TNXA (tenascin XA) is an extracellular matrix protein involved in cell adhesion and migration. The survival analysis of abnormally expressed TNXA in breast tissue indicates poor prognosis [84]. A low expression of ZBTB16 (zinc finger and BTB domain-containing protein 16) in BC has been associated with a poor prognosis and an increased risk of metastasis [85,86].

## 5. Conclusions

Artificial intelligence and machine learning approaches for the identification of novel diagnostic and prognostic gene signature biomarkers for breast cancer using microarray-based gene expression profiles were attempted. Initially, we identified a total of 355 DEGs via gene expression profiling of BC microarray data, and our artificial-intelligence-based strategy significantly reduced the number of genes needed for an effective evaluation of diagnostic and prognostic importance. As a result, two novel gene signatures were highlighted, (i) diagnostic nine-gene signature (*COL10A*, *S100P*, *ADAMTS5*, *WISP1*, *COMP*, *CXCL10*, *LYVE1*, *COL11A1*, and *INHBA*) and (ii) prognostic eight-gene signature (*CCNE2*, *NUSAP1*, *TPX2*, *S100P*, *ITM2A*, *LIFR*, *TNXA*, and *ZBTB16*), using machine learning algorithms and survival analysis. The results were confirmed via qPCR of BC samples, and validated by another set of ML methods to measure the model accuracy and precision. To the best of our knowledge, the identified diagnostic and prognostic gene signatures are novel and have clinical potential.

## Figures and Tables

**Figure 1 cancers-15-03237-f001:**
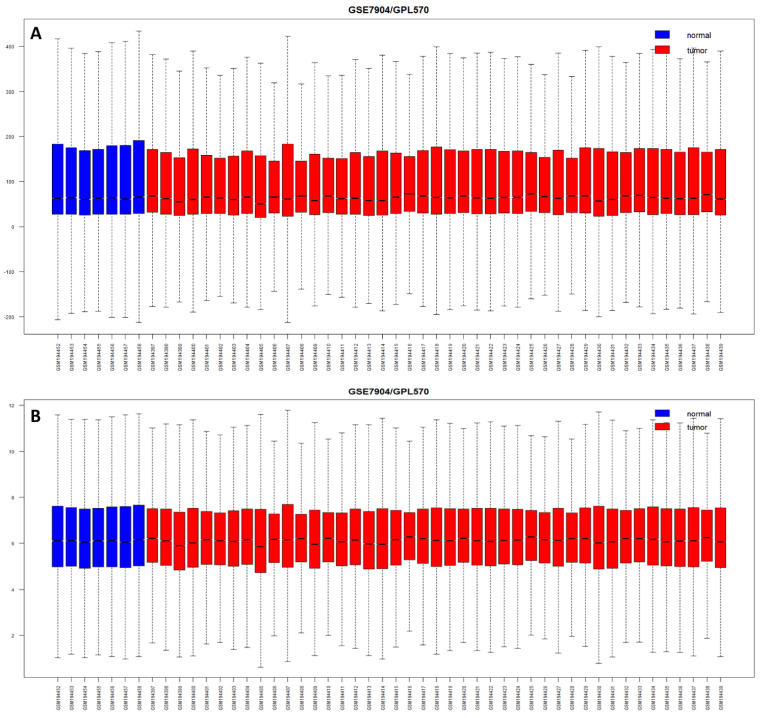
Boxplot showing the expression distribution for dataset GSE7904. (**A**) Raw (un-normalized) expression distribution with log2 scale in the range of −200 to 400. (**B**) Normalized intensities showing almost similar distributions of expression intensities, with the log2 scale in the range of 0 to 12.

**Figure 2 cancers-15-03237-f002:**
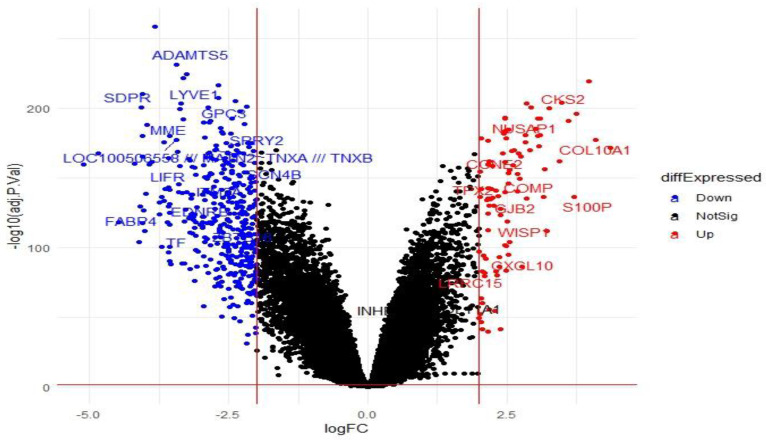
Volcano plot showing differentially expressed genes: (i) the majority were non-significant (black), (ii) upregulated DEGs (red), and (iii) downregulated DEGs (blue).

**Figure 3 cancers-15-03237-f003:**
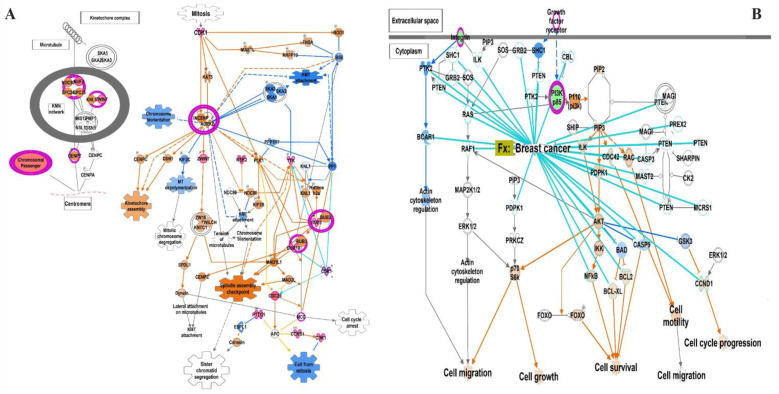
Canonical pathways derived using the IPA tool. (**A**) Kinetochore metaphase signaling pathway, (**B**) PTEN pathway overlapped with breast cancer associated genes.

**Figure 4 cancers-15-03237-f004:**
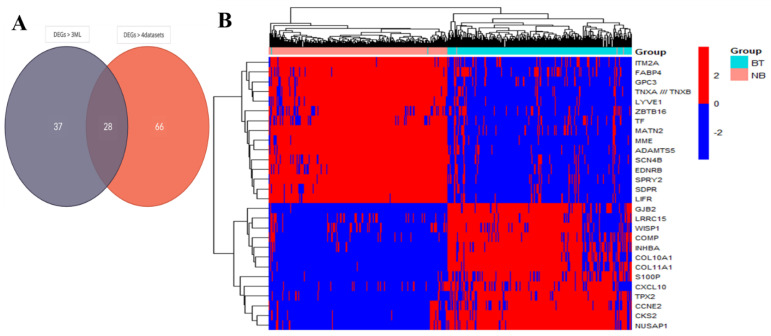
(**A**) Venn diagram showing 28 hub genes derived from the intersection of DEGs > 3 ML and DEGs > 4 datasets. (**B**) Unsupervised hierarchical clustering: heatmap of 701 samples, including 356 breast tumor (BT, cyan) and 345 normal breast (NB, pink) tissues, showing the gene expression pattern of 28 hub genes, including diagnostic and prognostic gene signatures. Upregulated genes are shown in red and downregulated genes are in blue.

**Figure 5 cancers-15-03237-f005:**
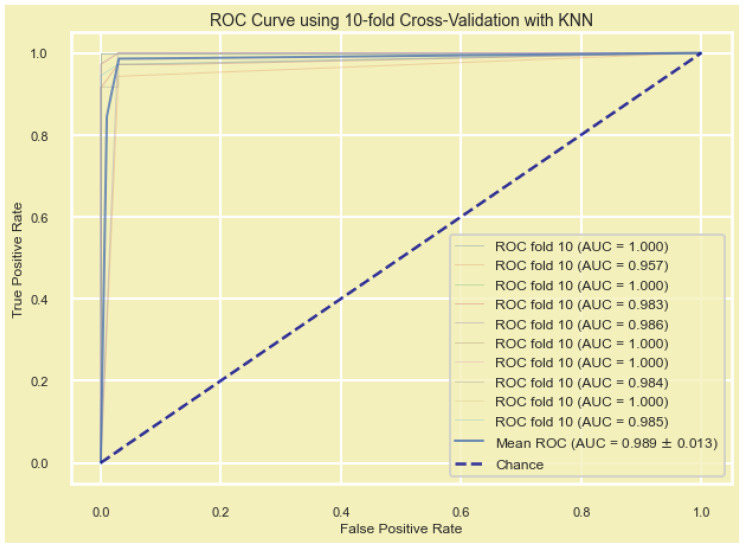
K-nearest neighbors (KNN)-based ML model for diagnostic gene signature showing the mean ROC (AUC 0.989 ± 0.013).

**Figure 6 cancers-15-03237-f006:**
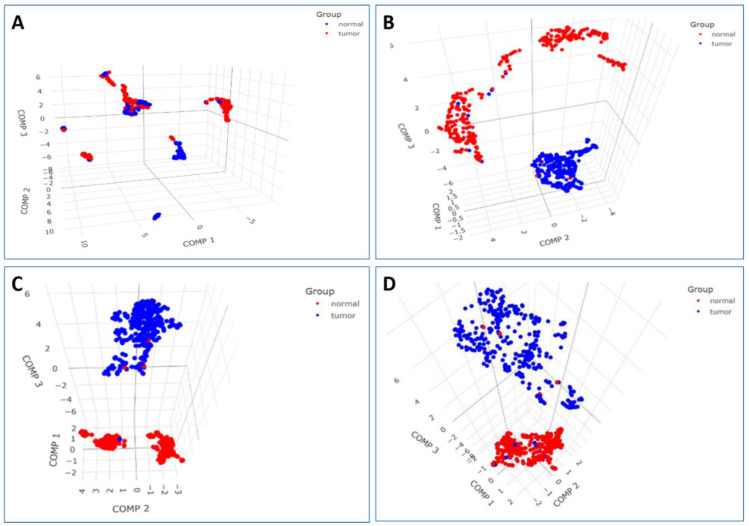
PCA plot showing an overall distribution of the samples (n = 701), including breast tumor (blue) and normal breast tissue (red) based on transcriptomics profiles: (**A**) 54,675 probes, (**B**) 355 DEGs, (**C**) 28 hub genes, and (**D**) diagnostic nine-gene signature.

**Figure 7 cancers-15-03237-f007:**
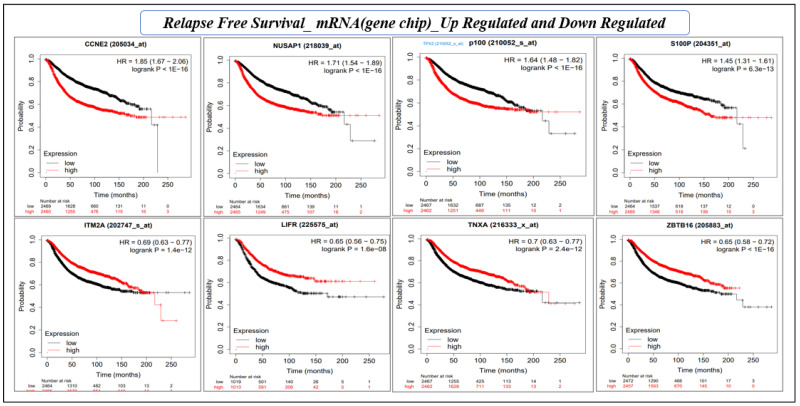
KM plot based on the relapse-free survival analysis of eight individual genes (mRNA, gene-chip) of prognostic gene signature. The *X*-axis and *Y*-axis represent time in months and the probability of the survival of patients, respectively. The impact of the high and low expression of the gene on patient survival is shown in red and black lines, respectively.

**Figure 8 cancers-15-03237-f008:**
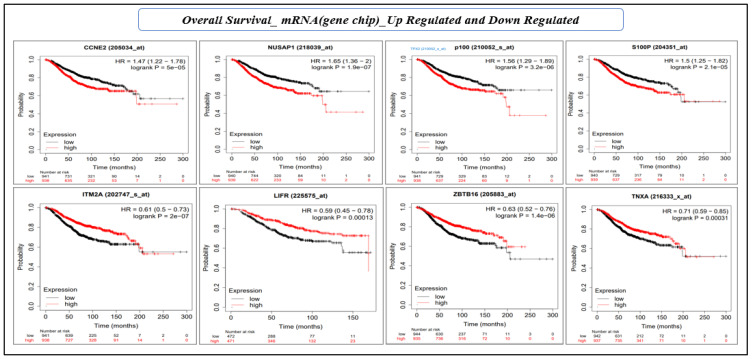
KM plot based on the overall survival analysis of eight individual genes (mRNA, gene-chip) of prognostic gene signature. The *X*-axis and *Y*-axis represent time in months and the probability of the survival of patients, respectively. The impact of the high and low expression of the gene on patient survival is shown in red and black lines, respectively.

**Figure 9 cancers-15-03237-f009:**
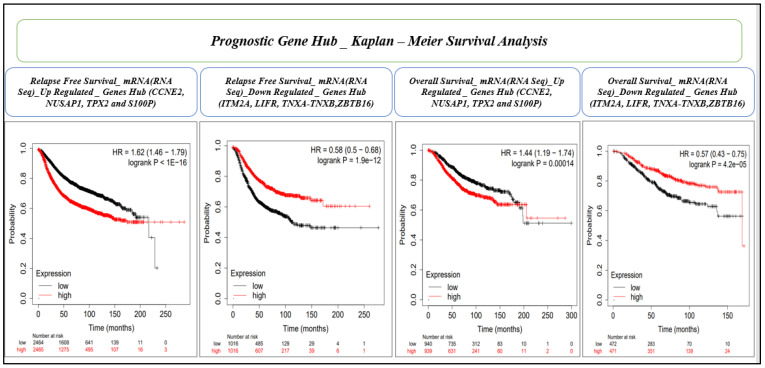
RFS and OS analyses and the validation of upregulated (*CCNE2, NUSAP1, TPX2*, and *S100P*), and downregulated (*ITM2A, LIFR, TNXA,* and *ZBTB16*) gene groups (mRNA, RNA seq) of the prognostic gene signature. The *X*-axis and *Y*-axis represent time in months and the probability of the survival of patients. The impact of the high and low expression of the gene on patient survival is shown in red and black lines, respectively.

**Figure 10 cancers-15-03237-f010:**
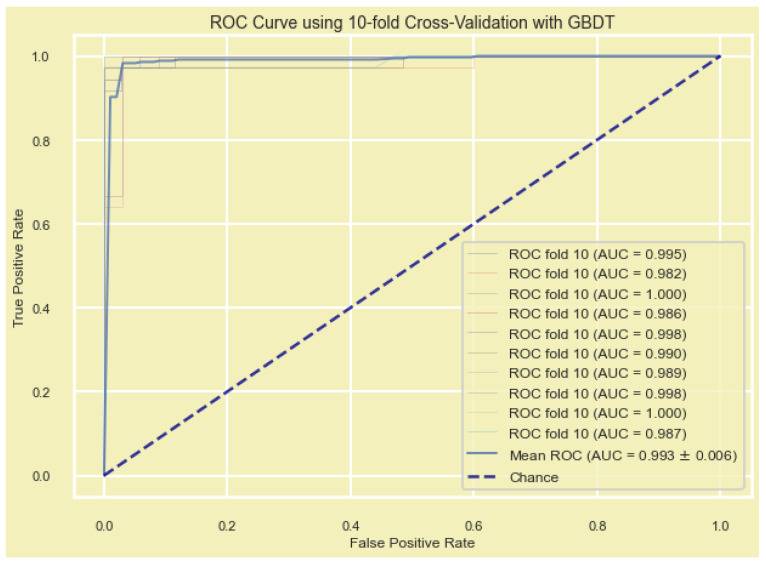
Gradient-boosting decision trees (GBDT) based on the ML model for the prognostic gene signature showing the mean ROC (AUC 0.993 ± 0.006).

**Figure 11 cancers-15-03237-f011:**
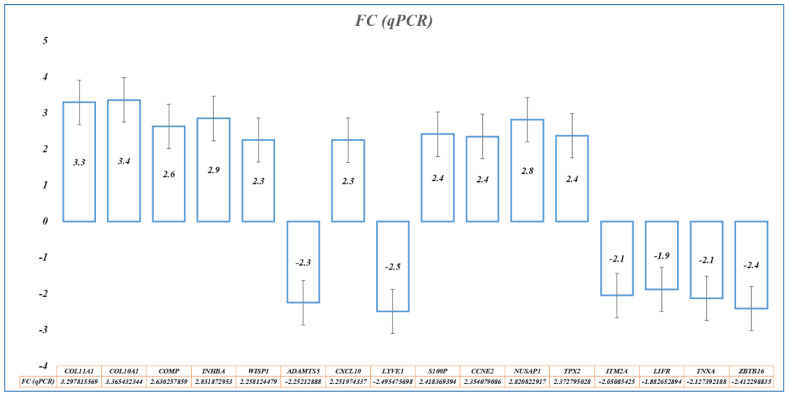
qRT-PCR results showing overexpression of *COL10A*, *S100P*, *WISP1*, *COMP*, *CXCL10*, *COL11A1*, *INHBA*; *CCNE2*, *NUSAP1*, *TPX2*, and *S100P* genes, and under-expression of *ADAMTS5*, *LYVE1*, *ITM2A*, *LIFR*, *TNXA*, and *ZBTB16* genes.

**Table 1 cancers-15-03237-t001:** GEO datasets from the GPL570 platform used for gene expression profiling.

Dataset	Title/Description	Normalization Methods	No. of Samples	Percentage of Cancer
GSE61304	Novel biomarker discovery for stratification and prognosis of breast cancer patients	MAS5 signal intensity	62 (58 breast tumor + 4 normal breast)	94%
GSE42568	Breast cancer gene expression analysis	Log2 GCRMA signal intensity	121 (104 breast tumor + 17 normal breast)	86%
GSE7904	Expression data from human breast tissue	RMA expression value	50 (43 breast tumor + 7 normal breast)	86%
GSE3744	Human breast tumor expression	GCRMA calculated signal intensity, log2 transformed	47 (40 breast tumor + 7 normal breast)	85%
GSE29431	Identifying breast cancer biomarkers	RMA expression values	66 (54 breast tumor + 12 normal breast)	82%
GSE26910	Stromal molecular signatures of breast and prostate cancer	Log2 RMA signal	12 (6 breast tumor + 6 normal breast)	50%
GSE31138	Identifying novel anti-angiogenic targets in human breast cancer	Log2 RMA signal	6 (3 breast tumor + 3 normal breast)	50%
GSE71053	Differential effect of surgical manipulation on gene expression in normal breast tissue and breast tumour tissue	Log2-normalized signal	18 (6 breast tumor + 12 normal breast)	33%
GSE10780	Proliferative genes dominate malignancy risk gene signature in histologically normal breast tissue	RMA expression value	185 (42 breast tumor + 143 normal breast)	23%
GSE30010	Expression data from breast samples of postmenopausal women	RMA expression value	107 (0 breast tumor + 107 normal breast)	0%
GSE111662	Whole breast tissue gene expression in comparison to expression in epithelial and stromal tissues	RMA expression values	27 (0 breast tumor + 27 normal breast)	0%
Total			701 (356 breast tumors + 345 normal breasts)	51%

**Table 2 cancers-15-03237-t002:** Top ten up- and downregulated differentially expressed genes from a total of 355 DEGs in breast cancer.

Gene Symbol	Gene Name	Log2FC	adj.*p*-Value	Decide Test
*COL11A1*	Collagen Type XI Alpha 1 Chain	4.36	1.69 × 10^−172^	Upregulated
*TOP2A*	DNA Topoisomerase II Alpha	3.96	4.16 × 10^−220^	Upregulated
*S100P*	S100 Calcium-Binding Protein P	3.70	3.57 × 10^−137^	Upregulated
*COL10A1*	Collagen Type X Alpha 1 Chain	3.59	7.47 × 10^−192^	Upregulated
*RRM2*	Ribonucleotide Reductase Regulatory Subunit M2	3.47	4.44 × 10^−205^	Upregulated
*CKS2*	CDC28 Protein Kinase Regulatory Subunit 2	3.26	8.66 × 10^−201^	Upregulated
*MMP1*	Matrix Metallopeptidase 1	3.21	7.95 × 10^−113^	Upregulated
*COMP*	Cartilage Oligomeric Matrix Protein	3.15	6.27 × 10^−137^	Upregulated
*NUSAP1*	Nucleolar And Spindle-Associated Protein 1	3.08	9.67 × 10^−194^	Upregulated
*ANLN*	Anillin, Actin-Binding Protein	3.07	2.42 × 10^−173^	Upregulated
*ADH1B*	Alcohol Dehydrogenase 1B (Class I), Beta Polypeptide	−4.84	2.10 × 10^−168^	Downregulated
*ADIPOQ*	Adiponectin, C1Q And Collagen Domain Containing	−4.47	6.08 × 10^−119^	Downregulated
*PLIN1*	Perilipin 1	−4.20	8.42 × 10^−161^	Downregulated
*LEP*	Leptin	−4.11	7.20 × 10^−105^	Downregulated
*LPL*	Lipoprotein Lipase	−4.09	2.35 × 10^−130^	Downregulated
*SDPR*	Serum Deprivation Response	−4.06	1.34 × 10^−201^	Downregulated
*RBP4*	Retinol Binding Protein 4, Plasma	−4.06	4.37 × 10^−118^	Downregulated
*C2orf40*	Chromosome 2 Open Reading Frame 40	−4.05	3.15 × 10^−211^	Downregulated
*ABCA8*	Atp-Binding Cassette Subfamily A Member 8	−4.05	4.21 × 10^−166^	Downregulated
*NTRK2*	Neurotrophic Tyrosine Kinase, Receptor, Type 2	−4.04	4.78 × 10^−181^	Downregulated

**Table 3 cancers-15-03237-t003:** Gene ontology of biological processes derived from the enrichment of differentially expressed genes in breast cancer.

Gene Set	Description	Gene Set Size	Expect Values	Overlap Value	Enrichment Ratio	FDR
GO:0031012	Extracellular matrix	487	09.48	35	3.69	3.31 × 10^−8^
GO:0051301	Cell division	576	11.21	41	3.65	2.69 × 10^−9^
GO:1903047	Mitotic cell cycle process	780	15.18	46	3.02	2.62 × 10^−8^
GO:0016477	Cell migration	1352	26.32	68	2.58	1.45 × 10^−9^
GO:0042127	Regulation of cell proliferation	1535	29.88	72	2.40	3.75 × 10^−9^
GO:0048870	Cell motility	1493	29.06	69	2.37	1.64 × 10^−8^
GO:0051674	Localization of cell	1493	29.06	69	2.37	1.64 × 10^−8^
GO:0009719	Response to endogenous stimulus	1574	30.64	71	2.31	2.01 × 10^−8^
GO:0008283	Cell proliferation	1953	38.02	87	2.28	6.20 × 10^−10^
GO:0009888	Tissue development	1814	35.31	77	2.18	3.31 × 10^−8^

**Table 4 cancers-15-03237-t004:** Machine learning methods for the 10-fold cross-validation of the diagnostic nine-gene signature.

ML Model	Mean AUC	Mean ACC	Mean Precision	Mean Recall	Mean F1
KNN	0.989	0.981	0.983	0.980	0.982
GBDT	0.995	0.973	0.970	0.978	0.973
AdaBoost	0.992	0.974	0.972	0.977	0.975
XGBoost	0.994	0.971	0.969	0.975	0.972
MLP	0.975	0.960	0.961	0.961	0.960

**Table 5 cancers-15-03237-t005:** mRNA (gene chip) and the relapse-free survival analysis of 28 hub genes, with the measured hazard ratio (HR), confidence interval (CI), and log-rank *p*-value.

Gene Symbol	Probe_IDs	HR	CI	Log-Rank *p*-Value	Decision (Log-Rank *p*-Value)
** *ADAMTS5* **	219935_at	0.9	0.77–0.94	1.50 × 10^−3^	Significant
** *CCNE2* **	205034_at	1.9	1.67–2.06	1.00 × 10^−16^	Significant
** *CKS2* **	204170_s_at	1.7	1.51–1.85	1.00 × 10^−16^	Significant
** *CXCL10* **	204533_at	1.2	1.12–1.37	4.40 × 10^−5^	Significant
** *EDNRB* **	206701_x_at	0.8	0.69–0.85	2.20 × 10^−7^	Significant
** *FABP4* **	203980_at	0.9	0.81–0.99	2.58 × 10^−2^	Significant
** *GPC3* **	209220_at	0.8	0.76–0.92	5.00 × 10^−4^	Significant
** *ITM2A* **	202747_s_at	0.7	0.63–0.77	1.40 × 10^−12^	Significant
** *LIFR* **	225575_at	0.7	0.56–0.75	1.60 × 10^−8^	Significant
** *MATN2* **	202350_s_at	0.9	0.78–0.95	3.30 × 10^−3^	Significant
** *LYVE1* **	219059_s_at	0.9	0.81–0.99	3.78 × 10^−2^	Significant
** *NUSAP1* **	218039_at	1.7	1.54–1.89	1.00 × 10^−16^	Significant
** *SCN4B* **	236359_at	0.6	0.55–0.75	1.00 × 10^−8^	Significant
** *SDPR* **	222717_at	0.7	0.57–0.77	9.70 × 10^−8^	Significant
** *SPRY2* **	204011_at	0.9	0.79–0.97	1.02 × 10^−2^	Significant
** *TF* **	214063_s_at	0.9	0.78–0.96	5.20 × 10^−3^	Significant
** *TNXA* **	216333_x_at	0.7	0.63–0.77	2.40 × 10^−12^	Significant
** *TPX2* **	210052_s_at	1.6	1.48–1.82	1.00 × 10^−16^	Significant
** *WISP1* **	229802_at	0.8	0.64–0.87	1.00 × 10^−4^	Significant
** *ZBTB16* **	205883_at	0.7	0.58–0.72	1.00 × 10^−16^	Significant
** *COL11A1* **	37892_at	1.2	1.12–1.38	2.30 × 10^−5^	Significant
** *INHBA* **	210511_s_at	1.2	1.06–1.3	1.70 × 10^−3^	Significant
** *S100P* **	204351_at	1.5	1.31–1.61	6.30 × 10^−3^	Significant
** *COL10A1* **	205941_s_at	1.0	0.88–1.08	6.60 × 10^−1^	Insignificant
** *COMP* **	205713_s_at	0.9	0.85–1.04	2.53 × 10^−1^	Insignificant
** *GJB2* **	223278_at	1.0	0.88–1.19	7.89 × 10^−1^	Insignificant
** *LRRC15* **	213909_at	0.9	0.82–1.01	7.16 × 10^−2^	Insignificant
** *MME* **	203435_s_at	1.1	0.98–1.2	1.28 × 10^−1^	Insignificant

**Table 6 cancers-15-03237-t006:** mRNA (gene chip) and the overall survival analysis of 28 hub genes, with the measured hazard ratio (HR), confidence interval (CI), and log-rank *p*-value.

Gene SymboL	Probe_IDs	HR	CI	Log-Rank *p*-Value	Decision (Log-Rank *p*-Value)
** *CCNE2* **	205034_at	1.47	1.22–1.78	5.00 × 10^−5^	Significant
** *CKS2* **	204170_s_at	1.32	1.09–1.59	3.70 × 10^−3^	Significant
** *ITM2A* **	202747_s_at	0.61	0.5–0.73	2.00 × 10^−7^	Significant
** *LIFR* **	225575_at	0.59	0.45–0.78	1.00 × 10^−4^	Significant
** *NUSAP1* **	218039_at	1.65	1.36–2	1.90 × 10^−7^	Significant
** *SDPR* **	222717_at	0.70	0.53–0.92	8.90 × 10^−3^	Significant
** *TNXA* **	216333_x_at	0.71	0.59–0.85	3.00 × 10^−4^	Significant
** *TPX2* **	210052_s_at	1.56	1.29–1.89	3.20 × 10^−6^	Significant
** *ZBTB16* **	205883_at	0.63	0.52–0.76	1.40 × 10^−6^	Significant
** *S100P* **	204351_at	1.50	1.25–1.82	2.10 × 10^−5^	Significant
** *ADAMTS5* **	219935_at	0.85	0.7–1.02	8.00 × 10^−2^	Insignificant
** *COL10A1* **	205941_s_at	0.96	0.79–1.15	6.38 × 10^−1^	Insignificant
** *COMP* **	205713_s_at	1.06	0.88–1.27	5.67 × 10^−1^	Insignificant
** *CXCL10* **	204533_at	0.9	0.75–1.09	2.98 × 10^−1^	Insignificant
** *EDNRB* **	206701_x_at	0.88	0.73–1.06	1.85 × 10^−1^	Insignificant
** *FABP4* **	203980_at	0.84	0.7–1.02	7.78 × 10^−2^	Insignificant
** *GJB2* **	223278_at	1.18	0.9–1.54	2.29 × 10^−1^	Insignificant
** *GPC3* **	209220_at	0.84	0.7–1.02	7.47 × 10^−2^	Insignificant
** *MATN2* **	202350_s_at	0.85	0.7–1.02	8.10 × 10^−2^	Insignificant
** *LRRC15* **	213909_at	0.87	0.72–1.04	1.30 × 10^−1^	Insignificant
** *LYVE1* **	219059_s_at	1.04	0.86–1.25	6.90 × 10^−1^	Insignificant
** *MME* **	203435_s_at	0.83	0.69–1.01	5.95 × 10^−2^	Insignificant
** *SCN4B* **	236359_at	0.83	0.63–1.08	1.68 × 10^−1^	Insignificant
** *SPRY2* **	204011_at	0.89	0.74–1.08	2.36 × 10^−1^	Insignificant
** *TF* **	214063_s_at	0.99	0.82–1.19	9.08 × 10^−1^	Insignificant
** *WISP1* **	229802_at	0.79	0.6–1.03	8.33 × 10^−2^	Insignificant
** *COL11A1* **	37892_at	1.12	0.93–1.35	2.37 × 10^−1^	Insignificant
** *INHBA* **	210511_s_at	1.12	0.93–1.36	0.2212	Insignificant
** *CKS2* **	204170_s_at	1.32	1.09–1.59	0.0612	Insignificant
** *SDPR* **	222717_at	0.7	0.53–0.92	0.0628	Insignificant

**Table 7 cancers-15-03237-t007:** mRNA (RNA seq) based on the relapse-free survival and overall survival analyses of eight prognostic gene hubs collectively measuring the hazard ratio (HR), confidence interval (CI), and log-rank *p*-value.

Survival Type	Gene Hub	HR	CI	Log-Rank *p*-Value	Decision by Log-Rank *p*-Value	Expression
RFS	*CCNE2, NUSAP1, TPX2, S100P*	1.62	1.46–1.79	1.00 × 10^−16^	Significant	Upregulated
RFS	*ITM2A, LIFR, TNXA-TNXB, ZBTB16*	0.58	0.50–0.68	1.90 × 10^−12^	Significant	Downregulated
OS	*CCNE2, NUSAP1, TPX2, S100P*	1.44	1.19–1.74	0.00014	Significant	Upregulated
OS	*ITM2A, LIFR, TNXA-TNXB, ZBTB16*	0.57	0.43–0.75	4.20 × 10^−5^	Significant	Downregulated

**Table 8 cancers-15-03237-t008:** Machine learning model for the 10-fold cross-validation of the prognostic eight-gene signature.

ML Model	Mean_AUC	Mean_ACC	Mean_Precision	Mean_Recall	Mean_F1
GBDT	0.993	0.980	0.983	0.977	0.980
XGBoost	0.992	0.976	0.981	0.972	0.976
AdaBoost	0.987	0.967	0.965	0.972	0.968
KNN	0.985	0.979	0.978	0.980	0.979
MLP	0.979	0.966	0.975	0.958	0.966

**Table 9 cancers-15-03237-t009:** Expression of gene signatures in microarray and qRT-PCR. The results of qRT-PCR are represented by the quantitative expression (Rq) and fold-change (FC), along with the standard deviation (StdDev) and *p*-values.

Gene	Microarray		qRT-PCR			
	FC	adj.*p*-Value	Rq	FC	StdDev	*p*-Value
**Expression of Diagnostic Gene Signature**						
COL11A1	3.907519	5.1 × 10^−163^	9.834254	3.297816	0.64495	3.47 × 10^−7^
COL10A1	3.842333	2.1 × 10^−178^	10.30614	3.365432	0.660567	1.84 × 10^−8^
S100P	3.701498	3.6 × 10^−137^	5.345665	2.418369	0.995349	2.73 × 10^−11^
COMP	3.150415	6.3 × 10^−137^	6.191366	2.630258	0.425433	5.81 × 10^−15^
INHBA	3.042628	2.6 × 10^−157^	7.21937	2.851873	0.901961	1.97 × 10^−8^
WISP1	2.551547	6 × 10^−105^	4.783692	2.258124	0.461439	2.9 × 10^−8^
ADAMTS5	−3.13169	3.3 × 10^−184^	0.209914	−2.25213	0.511629	2.88 × 10^−9^
CXCL10	2.530934	2.16 × 10^−95^	4.763343	2.251974	0.86944	2.16 × 10^−5^
LYVE1	−3.14204	1.2 × 10^−142^	0.177332	−2.49548	0.877592	1.73 × 10^−6^
**Expression of Prognostic Gene Signature**						
CCNE2	2.530327	3.7 × 10^−154^	5.112678	2.354079	0.392123	3.1 × 10^−15^
NUSAP1	2.732299	2.4 × 10^−124^	7.065653	2.820823	0.9127	3.81 × 10^−10^
TPX2	2.145025	5.6 × 10^−135^	5.179436	2.372795	0.432888	4.72 × 10^−10^
ITM2A	−2.54576	9.1 × 10^−149^	0.241341	−2.05085	0.683145	4.21 × 10^−8^
LIFR	−3.0494	5.6 × 10^−159^	0.271185	−1.88265	0.853309	1.78 × 10^−6^
TNXA	−2.54523	1.9 × 10^−129^	0.228871	−2.12739	0.736176	7.89 × 10^−7^
ZBTB16	−2.4943	1.12 × 10^−115^	0.187856	−2.4123	0.567998	3.32 × 10^−9^
S100P	3.701498	3.6 × 10^−137^	5.345665	2.418369	0.995349	2.73 × 10^−11^

## Data Availability

All data are publicly available, and their web links are mentioned in the Methods Section. However, additional information can be provided by the authors upon reasonable request.

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
