# Peer review of "Identification of Novel Diagnostic and Prognostic Gene Signature Biomarkers for Breast Cancer Using Artificial Intelligence and Machine Learning Assisted Transcriptomics Analysis"

_cancers, 2023, doi:10.3390/cancers15123237_

Round 1
Reviewer 1 Report
It was evaluated the article “Identification of Novel Diagnostic and Prognostic Gene Signature Biomarkers for Breast Cancer Using Artificial Intelligence and Machine Learning Assisted Transcriptomic Analysis”.
The goal was “to identify the valuable gene-signature model based on differentially expressed genes (DEGs) for BC diagnosis and prognosis”.
The article is extremely interesting and well done. It involves an important topic and the current AI approach.
Some minor concerns were raised.
- 701 samples were used. Where are the sample size calculation and power analysis to show the significance of this sample size?
- Where are the IRB approval?
Author Response
Response to reviewer 1:
Comments and Suggestions for Authors
Comment 1: It was evaluated the article “Identification of Novel Diagnostic and Prognostic Gene Signature Biomarkers for Breast Cancer Using Artificial Intelligence and Machine Learning Assisted Transcriptomic Analysis”. The goal was “to identify the valuable gene-signature model based on differentially expressed genes (DEGs) for BC diagnosis and prognosis”. The article is extremely interesting and well done. It involves an important topic and the current AI approach.
Response 1: We appreciate your effort to review this article and share your concerns.
Comment 2: 701 samples were used. Where are the sample size calculation and power analysis to show the significance of this sample size?
Response 2: Yes, a larger sample size gives greater analysis power because it constricts the distribution of the test statistic. We have incorporated following information to the revised manuscript.
We employed LIMMA, which incorporates an empirical Bayes estimate, to adjust the standard deviation in the t-test denominator by considering the distribution of all standard deviations. Wang et al. (2021) conducted a study demonstrating the superior performance of the moderated t-test when the sample size was ≥40. Our study, with a sample size of 701 comprising 356 breast tumors and 345 normal breast samples, exceeds the required 95% power of the test, and with nearly equal representation of test (tumor) and control (normal) groups mitigates biases in machine learning-based data analysis and enhances the accuracy of the model and predicted biomarkers.
Wang, G., Muschelli, J., & Lindquist, M. A. (2021). Moderated T-tests for group-level fmri analysis. NeuroImage, 237, 118141. https://doi.org/10.1016/j.neuroimage.2021.118141
Comment 3: Where are the IRB approval?
Response 3: Thanks for pointing out the IRB approval issue, now it is added in the revised manuscript under “data sets and patients” section of materials and methods. “This study was approved by the university’s CEGMR bioethical committee (16-CEGMR-bioeth-2022) dated 13-10-2022, and we recruited patients for validation of potential biomarkers after getting their consents.”
Reviewer 2 Report
The paper is interesting and for some aspects innovative respect the existing literature. The topic is timely and it could be appreciated by the reader.
I suggest to interlace the reported approach with innovative image signal processing
I suggest to look at the following paper and to remark my idea.
Moreover the paper could complete the literature:
Image processing for medical diagnosis using CNN Authors Paolo Arena, Adriano Basile, Maide Bucolo, Luigi Fortuna Publication date 2003/1/21 Journal Nuclear Instruments and Methods in Physics Research Section A: Accelerators, Spectrometers, Detectors and Associated Equipment Volume 497 Issue 1 Pages 174-178 Publisher North-Holland The fusion of more techniques in data processing for cancer study must be a further instrument to get successful results.The english must be softly reviewed.
Author Response
Reviewer 2:
Comments and Suggestions for Authors
Comment 1: The paper is interesting and for some aspects innovative respect the existing literature. The topic is timely and it could be appreciated by the reader.
Response 1: We appreciate the effort of reviewer for valuable suggestions. Manuscript has been revised accordingly.
Comment 2: I suggest to interlace the reported approach with innovative image signal processing.
Response 2: It’s a good idea to include the image signal process but in the present article, the focus was to develop an AI/ML-based model and identify biomarkers using transcriptomic data.
Comment 3: I suggest to look at the following paper and to remark my idea. Moreover the paper could complete the literature: Image processing for medical diagnosis using CNN Authors Paolo Arena, Adriano Basile, Maide Bucolo, Luigi Fortuna Publication date 2003/1/21 Journal Nuclear Instruments and Methods in Physics Research Section A: Accelerators, Spectrometers, Detectors and Associated Equipment Volume 497 Issue 1 Pages 174-178 Publisher North-Holland The fusion of more techniques in data processing for cancer study must be a further instrument to get successful results.
Response 3: We included the suggestion as follows: “AI and ML techniques based on automated medical diagnosis are increasing gradually for clinical, pathological, and radiological reports. The fusion of multiple techniques in different types of data processing for cancer studies must be a further instrument to get successful results. Earlier convolution neural network approaches had been applied for image processing in medical diagnosis [58].
Paolo Arena, Adriano Basile, Maide Bucolo, Luigi Fortuna. Image processing for medical diagnosis using CNN. Nuclear Instruments and Methods in Physics Research Section A: Accelerators, Spectrometers, Detectors and Associated Equipment, Volume 497, Issue 1, 2003, Pages 174-178, https://doi.org/10.1016/S0168-9002(02)01908-3.
Comment 4: Comments on the Quality of English Language: The english must be softly reviewed.
Response 4: We did English language editing to remove any grammar or typo errors.